# The association between shame and substance use in young people: a systematic review

Masuma Rahim and Robert Patton

University of Surrey, United Kingdom

## ABSTRACT

**Background.** Shame has been associated with a range of maladaptive behaviours, including substance use. Young people may be particularly vulnerable to heightened shame sensitivity, and substance use is a significant problem amongst UK adolescents. Although there appears to be a relationship between shame and substance use, the direction of the relationship remains unclear.

**Aim.** The purpose of this study was to undertake a systematic review of the literature relating to shame and substance use in young people.

**Method.** Five electronic databases were searched for articles containing terms related to 'adolescence,' 'shame' and 'substance use.' Six articles were included in the final analyses.

**Results.** Adverse early experiences, particularly sexual abuse, predict shame-proneness, and substance use is a mechanism by which some individuals cope with negative feelings. In general, there is a dearth of literature investigating the shame-substance use relationship in adolescent samples. The available literature associates shame-proneness with poorer functioning and suggests that it may potentially lead to psychopathology and early-onset substance use. Scant attention has been paid to the cognitive and emotional processes implicated. Further research is required to ascertain the strength of the shame-substance use relationship in young people and to develop appropriate interventions for this population.

Corresponding author
Robert Patton,
r.patton@surrey.ac.uk

## INTRODUCTION

A significant body of research has investigated the effect of factors related to self-concept on substance use in young people. Self-esteem, impulsivity and shame have all been associated with alcohol and other drug use in young people but existing understanding of the association with shame in particular is limited. This paper will review current empirical and conceptual understanding of the association between shame and substance use in this population. It is of note that drug and alcohol use ranges from 'experimental' to 'harmful' and that some young people report 'non-problematic' or 'recreational' use (*Bauman & Phongsavan, 1999*; *Martin, Chung & Langenbucher, 2008*). The papers reviewed here use a variety of terms to describe use of alcohol and other drugs; consequently, the term 'substance use' refers to all levels of use throughout this paper.

Shame relates to global, negative feelings about the self (*Dearing, Stuewig & Tangney, 2005*) and has been described as an intense negative emotion which can result in feelings of inferiority and powerlessness (*Wicker, Payne & Morgan, 1983*). Shame can arise from a disparity between the ideal self and the actual real self, leading to feelings of inadequacy and disgust. Although it is generally associated with negative consequences, it is notable that shame is universally experienced and that moderate levels can be beneficial in enabling individuals to evaluate themselves and their actions and to moderate them appropriately (*Cook, 1987*; *Nathanson, 1987*; *Potter-Efron, 1987*). Much of the literature has sought to distinguish the concepts shame and guilt although, historically, it has failed to do so adequately (*Kugler & Jones, 1992*). More recently, it has been suggested that both emotions are essential to the experience of being human and that they can occur either independently or in tandem with each other (*Clark, 2012*). Both shame and guilt enable self-evaluation and serve to guide our interactions with others (*Tangney & Dearing, 2002*). Shame has been described as a 'failure of being,' or global self-condemnation, whilst guilt has been referred to as a 'failure of doing' (*Potter-Efron, 1987*). The former may result in feelings of inadequacy, deficiency and being 'exposed'; the latter is associated with an individual feeling 'wicked' and remorseful. Whilst vulnerability to shame can arise from a conflict between the ideal and actual self, vulnerability to guilt may result from conflict between the actual self and the 'should' self (*Moretti & Tory-Higgins, 1990*). It appears that shame is directed primarily at the self, whereas guilt addresses the particular act, and may be implicated in conformity to societal norms (*Quiles, Kinnunen & Bybee, 2002*). Shame-proneness is often internalized and has been associated with the development of psychopathology, whilst proneness to guilt, generally more overt, correlates with non-pathological, adaptive empathy (*Tangney, Wagner & Gramzow, 1992*). However, far from being a purely negative emotional state, feeling shameful warns the individual that their actions are socially unacceptable and may result in them being rejected by others. In order to avoid rejection, the individual seeks to find alternative ways of behaving (*Nathanson, 1987*). As such, shame is characterised by 'hiding' the self, whereas guilt may manifest itself in reparative behaviours. Consequently, similar situations can result in distinct responses, depending on the individual's attributional style (*Lewis, 2008*).

## The development of shame in children and young people

The development of shame is dependent on individual possessing sufficient cognitive capacity, having an awareness of social rules and expectations, and an understanding of their behaviour in comparison to those expectations, as well as adequate theory of mind (*Gilbert, 2002*; *Lewis, 2003*). As such, it is unlikely that shame develops before the age of two years (*Zahn-Waxler et al., 1992*).

Sexual abuse, insecure attachment styles and harsh parenting have all been associated with the development of shame in children (*Feiring et al., 2002*; *Gross & Hansen, 2000*; *Jeffrey & Jeffrey, 1991*; *Stuewig & McCloskey, 2005*). Shame may develop as a secondary consequence of early adversity; in families where factors such as parental substance use are implicated, children may develop more empathic attitudes as they try to minimise

**Peer**J

parental disturbance (*Lewis, 1995*). If they fail to do so, they are likely to blame themselves, developing a negative global attributional style and heightened proneness to shame (*Lewis, 1995*). Differences in the ways males and females are socialized can result in females developing a greater sense of responsibility and becoming more shame-prone than their male peers, despite experiencing similar early life events (*Lewis, 1995*; *Tangney, 1990*).

Shame is likely to manifest itself according to the child's developmental stage. Young children may experience shame as feelings of embarrassment and inferiority; coping with this negative affect by behaving in a controlling, critical or rageful manner (*Bradshaw, 2005*). They may also polarise themselves and others as being 'all good' or 'all bad'. During puberty shame can be experienced as limiting one's ability to form their identity and may serve to isolate the individual (*Bradshaw, 2005*). Self-evaluation is central to the development of shame in both younger children and adolescents (*Lewis, 1995*; *Lewis, 2003*; *Reimer, 1996*). Adolescence is conceptually characterised by the development of identity and by separation from caregivers (*Koepke & Denissen, 2012*) and It is often here that young people begin to develop meaningful peer relationships (*Allen & Land, 1999*). As their capacity for self-reflection and social perspective-taking develops, adolescents are more likely to compare themselves negatively to peers, (*Reimer, 1996*). Some young people may develop an enhanced vulnerability to feeling shameful, potentially resulting in the use of maladaptive coping strategies such as substance use and making them more vulnerable to depression, eating disorders and suicide (*Reimer, 1996*). There is some evidence that alcohol use demonstrates a stronger association with psychopathology than other drugs, such as marijuana (*Dearing, Stuewig & Tangney, 2005*).

## Substance use amongst adolescents

In the UK, 84% of 12–17 year-olds have drunk alcohol. Over a third are frequent drinkers and 15% have been involved in antisocial behaviour as a result of drinking (*Institute for Alcohol Studies, 2013*). The same report stated that two major reasons given for underage drinking were 'escapism' and 'gaining respect from peers'. With reference to illicit substances, 18% of 11–15 year-olds report having used drugs; 12% having done so in the past year. Cannabis is the most widely-used controlled drug, taken by 8% of 11–15 year-olds in 2010. Truanting and school suspension are highly correlated with regular drug use (*National Health Service Information Centre, 2011*), and those who begin using alcohol between the ages of 14 and 16 are four times as likely to develop alcohol-related problems in later life than those who begin drinking in their 20s (*Gunzerath et al., 2011*). Despite this, outcomes following intervention remain variable: although 23,000 Britons under the age of 18 accessed support for substance use in 2009/10, only one in three completed treatment (*National Health Service Information Centre, 2011*). Given that the primary motivators underlying substance use relate to coping with negative affect, it may be the case that interventions need to more effectively target these factors.

## The relationship between shame and substance use

In adults shame has been strongly implicated in behaviours which enable individuals to escape feelings of worthlessness and failure, such as binge-eating, sexual risk-taking and
substance use (*Adams & Robinson, 2001*; *Hayaki, Friedman & Brownell, 2002*; *Peñas-Lledó, Fernández & Waller, 2004*; *Talbot, Talbot & Tu, 2004*). Heightened shame significantly increases vulnerability to addictive behaviours, particularly substance use (*Cook, 1987*). Although several studies indicate that shame which arise from the stigma surrounding substance use may serve as a barrier to treatment (*Cook, 1987*; *Corrigan, Watson & Miller, 2006*; *Luoma et al., 2007*), it has been suggested that this stigma is more pronounced amongst certain groups. In particular, females seeking treatment for substance use and related problems may face greater stigma than males, often risking the breakdown of intimate relationships, as well as the removal of their children (*Blume, 1990*; *O'Connor et al., 1994*; *Reed, 1987*). Consequently, females who enter treatment programs often experience enhanced shame and guilt compared to their male counterparts (*Mason, 1991*).

### Rationale for the current review

As described, there is some evidence that shame arises from early adversity and that it is correlated with a range of maladaptive behaviours, including substance use. Substance use is a significant problem amongst young people but little research has investigated the impact of shame on alcohol and illicit drug use in this population. This paper presents a systematic review of the existing evidence investigating the relationship between shame and substance use in young people in order to identify the areas that substance use interventions might focus on in this population.

## SEARCH STRATEGY

Five electronic databases were searched (PsycArticles, PsycInfo, Medline, Web of Science and PubMed) for English-language articles published in peer-reviewed journals for all periods up to, and including, January 2012. Articles were searched for using terms related to 'adolescence' (e.g., 'adolescen\*,' 'teen\*,' 'child\*,' 'juvenile\*,' 'youth\*'), 'shame' (e.g., 'shame\*') and 'substance use' (e.g., 'substanc\*,' 'drug\*,' 'alcohol\*,' 'illicit\*,' 'drink\*'). The use of '\*' denotes truncated search terms in order to increase the number of records retrieved. For the purposes of this review, 'adolescent' refers to individuals aged 11–19. Where databases could be searched by topic, 'psychology' was specified. After the removal of duplicate records, the search yielded 220 results. Following review of the abstracts, articles were excluded if they were unavailable in English ($n = 9$) or if they focused on unrelated physical or psychological health problems ($n = 122$). Of the remaining 89 articles, 61 were excluded as they did not contribute to the literature relating to the relationship between shame and substance use (e.g., 19 papers focused on shame arising from parental substance use). The remaining articles ($n = 28$) were read to ensure that they included a research question and outcome measures. Seven papers exclusively reviewed the literature, but were not empirical studies. Another five discussed cases and process issues in therapy but did not relate specifically to shame as either a contributing factor or a consequence of substance use. Given that these papers did not contribute to the scientific understanding of the relationship between shame and substance use, they were excluded. A further ten papers were excluded as they used samples of children (pre-teen) or adults (post-adolescent). The final analyses included six empirical papers.

## RESULTS

Details of the articles reviewed can be seen in Table 1. All of the papers had been published between 1989 and 2012. Four of the studies were quantitative in methodology; two used a qualitative design. Amongst the quantitative studies, sample sizes ranged from 97–816. The qualitative studies employed 12 and 597 participants, respectively. One paper considered the development of shame in young people in treatment for substance use; five investigated the relationship between shame and substance use.

### The role of sexual abuse in the development of shame

Of the six studies reviewed, one investigated factors resulting in heightened shame in later life (*Edwall, Hoffmann & Harrison, 1989*), identifying sexual abuse, including incest, as a predictor of shame-proneness. In their qualitative study of 597 adolescent girls, it was found that 35% of inpatients receiving treatment for substance use reported previous sexual abuse. Sexual abuse was highly correlated with both a history of physical abuse and with having attempted in the previous year. Sexual abuse was also strongly associated with feelings of shame, particularly amongst those who had experienced only extra familial abuse ($p < 0.001$). The authors concluded that adolescent girls with a history of abuse may internalize adverse experiences and construct an image of themselves as 'bad,' making them vulnerable to suicidal ideation and the development of mental health problems. However, the categorisation of those who had or had not been abused was problematic; 58 girls who denied having been sexually abused during interviews with the researchers were classified by their therapists as having reported sexual abuse in therapy sessions, and excluded from analyses. Additionally, the researchers made no use of standardised measures, and thus the severity of the shame experienced cannot be assessed. No attempt was made to ascertain the duration or nature of the sexual abuse and only limited information pertaining to the course of the participants' use of substances was available.

### The relationship between shame and substance use

Two of the studies included in this review found significant associations between shame-related affect and maladaptive behaviours, including substance use (*Abramowitz & Berenbaum, 2007*; *Dearing, Stuewig & Tangney, 2005*). *Abramowitz & Berenbaum*'s (*2007*) study found that the desire to enhance positive affect was a strong motivator of alcohol use, and that shame reliably predicted a range of 'impulsive-compulsive' (IC) behaviours, including substance use, sexual activity, playing video games and obsessive-compulsive-type behaviours such as cleaning. Shame was most strongly associated with substance use (correlation $= 0.16$; $p < 0.05$). It is of note, however, that their data were based on retrospective accounts of behaviours the participants had engaged in during the past three months and that, although the associations reported were statistically significant, causality could not be ascertained. In addition, the sample was aged 16–30 and, as the authors note, many impulsive and compulsive behaviours diminish with age.

Carrying out semi-structured interviews with college students, *Lashbrook (2000)* found that the desire to avoid ridicule, isolation and feelings of inadequacy was key in alcohol

Rahim and Patton (2015), *PeerJ*, DOI 10.7717/peerj.737

**Table 1** Included articles.

| Authors & title | Year | Research aim | Measures | Sample | n | Findings | Evaluation |
|---|---|---|---|---|---|---|---|
| Abramowitz, A, & Berenbaum, H. Emotional triggers and their relation to impulsive and compulsive psychopathology. | 2007 | Emotional triggers as a predictor of impulsive-compulsive behaviours | BIS-11, OCI | College Students | 189 | Anger & shame predict I-C pathology | Correlational study |
| Dearing, RL, Stuewig, J, & Tangney, JP. On the importance of distinguishing shame from guilt: relations to problematic alcohol and drug use. | 2005 | Clarifying the role of shame and guilt in substance use | MCMI & TOSCA | Students & inmates | 816 | Shame correlates with substance use in both samples | Correlation |
| Edwall, GE, Hoffmann, NG, & Harrison, PA. Psychological correlates of sexual abuse in adolescent girls in chemical dependency treatment. | 1989 | Psychopathology & self-concept in victims of sexual use | Interviews by counsellors | Adolescent females | 597 | Sexual abuse & shame are common in substance use | Limited data re: severity of abuse |
| Lashbrook, JT. Fitting in: exploring the emotional dimension of adolescent peer pressure. | 2000 | Emotions and conformity | Qualitative | College students | 12 | Facets of shame motivate individuals to drink alcohol with peers | Small sample; Retrospective |
| Quiles, ZN, Kinnunen, T, & Bybee, J. Aspects of guilt and self-reported substance use in adolescence. | 2002 | The relationship between guilt and adolescent substance use | TOSCA, GI, MFCGI, PFQ2 | Students | 230 | Substance users have weaker internalisation of societal standards | Retrospective, self-report data Focus on guilt |
| Rosenkranz, SE, Henderson, JL, Muller, RT & Goodman, IR. Motivation and maltreatment history among youth entering substance abuse treatment. | 2012 | The relationship between maltreatment and motivation to change | SOCRATES, TEQ, TAQ, AUDIT, DAST, PSS | 16–24 year-old substance abusers | 188 | Shames is associated with substance use | 89% positive response rate self-reported maltreatment |

use. Despite the participants in this study not using terms such as 'shame' explicitly, the literature suggests that ridicule, isolation and inadequacy are closely linked to constructs of shame-proneness (*Cook, 1987*; *Potter-Efron, 1987*; *Wicker, Payne & Morgan, 1983*). It is possible that low self-esteem was implicated in these negative emotions, although this was not investigated within the study. However, findings reported by *Dielman et al. (1987)* provide support for the suggestion that low self-esteem is associated with vulnerability to peer pressure and increased substance use.

*Dearing, Stuewig & Tangney (2005)* conducted three studies to test the relationships between shame- and guilt-proneness amongst undergraduate students and prisoners, hypothesising that the former would correlate positively with substance use, whilst the opposite would be true for guilt-proneness. Of their undergraduate sample, 7.3% reported symptoms consistent with problematic alcohol use and 15.4% indicated symptoms of problematic drug use. Shame was positively correlated with alcohol problems but not with problematic drug use. In contrast, guilt was found to correlate negatively with both drug and alcohol problems; findings which supported those presented by *Quiles, Kinnunen & Bybee (2002)*. The authors concluded that although shame-proneness showed some association with problematic drug use, stronger effects were observed with alcohol use. The shame-substance use relationship appears consistent in a range of populations. The authors suggested that alcohol and other drugs may be used in order to cope with negative affect such as shame but noted that use of substances may in itself result in additional shame.

The majority of the studies included in this review indicate that vulnerability to shame is associated with increased drug and alcohol use in young people. Some research suggests that feelings of shame can arise as a consequence of using substances (*Arentzen, 1978*; *Blume, 1990*; *Cook, 1987*; *Corrigan, Watson & Miller, 2006*; *Fossum & Mason, 1986*; *Luoma et al., 2007*; *O'Connor et al., 1994*; *Reed, 1987*), and, of the papers reviewed here, one suggests that, amongst those who already use substances problematically, shame may have a positive impact by increasing motivation to seek treatment. In their study of 188 16–24 year-olds entering treatment for moderately-problematic substance use, *Rosenkranz et al. (2012)* found that those individuals who reported greater shame-proneness were more likely to recognize their substance use as problematic and to seek treatment. Some research suggests that these individuals demonstrate superior treatment outcomes (*Williams et al., 2008*), but their data were subject to disclosure biases. Similarly, *Rosenkranz et al. (2012)* used a measure of treatment motivation which conflates proneness to shame with motivation to seek treatment, and which included items (e.g., If I remain in treatment it will probably be because I'll feel very bad about myself if I don't) open to being interpreted as either 'shame' or 'guilt' by participants.

## DISCUSSION

### Summary of findings

This paper sought to review the literature relating to shame and substance use in young people. Despite the search strategy specifying 'adolescent,' and variations thereof, only five papers used samples which investigated teenagers and young adults. An additional study

carried out by *Quiles, Kinnunen & Bybee (2002)*, used a sample aged ≤27 but excluded all those aged 22–27 ($n = 17$) from their analyses; thus, the paper was included in this review. Amongst the remainder of the studies participants ranged in age from 7 to 80. It appears that there is a significant absence of research into the association between shame and substance use amongst young people and this paucity limits the extent to which theoretical or empirical conclusions can be drawn. It appears that early maltreatment and neglect can result in heightened shame-proneness, possibly as a result of adverse experiences being internalized, and that greater maladjustment results from more severe adversity. Although shame arising from maladaptive early experiences has been found to correlate significantly with substance-using behaviours, it can also motivate individuals to seek and engage in treatment. It is concluded from the evidence reviewed that it is shame which is most heavily implicated in these mechanisms, not guilt.

The literature suggests that shame-proneness is generally associated with negative outcomes such as poorer functioning, psychopathology and early-onset substance use Adolescents are more likely to compare themselves negatively to peers (*Reimer, 1996*) and those who develop a heightened proneness to shame may be more likely to utilise coping strategies such as criminal or risk-taking behaviours in an attempt to gain peer acceptance (*Adams & Robinson, 2001*; *Arnett, 1995*; *Hayaki, Friedman & Brownell, 2002*; *Peñas-Lledó, Fernández & Waller, 2004*; *Talbot, Talbot & Tu, 2004*).

Whilst there is some indication that enhanced shame results in a greater number of maladaptive behaviours (*Cook, 1987*), it is also suggested that shame-proneness affects males and females differently. Based upon the evidence reviewed, it is tentatively speculated that shame in females results in behaviours which harm the self, such as eating disorders, whereas males tend externalise negative self-image and may act in an antisocial manner. It is suggested that, if shame encompasses negative affect and symptoms typically observed in depression, it is to be expected that young people who demonstrate a tendency towards feeling shame would also score poorly on measures of self-worth and self-esteem.

The studies included in this review were conducted using a wide range of outcome measures and methodological designs; each of which demonstrated both strengths and limitations. For the most part, researchers made efforts to distinguish between shame and guilt, a key methodological requirement, given the conceptual overlap in these variables. Nevertheless, it is of note that each of the studies reviewed conceptualised shame in distinct ways and investigated different facets of substance use, further limiting the extent to which overarching conclusions can be drawn.

## Gaps in the literature

In addition to the dearth of literature focusing exclusively on adolescent populations, the majority of studies have failed to address use of substances in a discrete manner. Although some studies considered alcohol and other drug use separately (*Dearing, Stuewig & Tangney, 2005*; *Quiles, Kinnunen & Bybee, 2002*), not all did so. It cannot be presumed *a priori* that all addictive behaviours are a product of the same mechanisms and this requires further investigation.

More fundamentally, although there is some evidence of an association between shame and substance use, there has been little focus on the cognitive and emotional processes which mediate this relationship. Shame is associated with substance use, which has a profound impact on both society and the individual; as such, future work should aim to identify salient risk factors and develop effective treatments. This review has not included studies which aimed to treat substance use per se; although two studies did recruit participants engaged in treatment programs (*Edwall, Hoffmann & Harrison, 1989*; *Rosenkranz et al., 2012*) and there is some indication that reducing shame is integral to positive treatment outcomes. Some research suggests that particular factors related to shame, specifically 'fragility and lack of control' and 'loneliness and emptiness,' appear to be associated with addiction (*Cook, 1987*), and future work should investigate the specific antecedents and maintenance processes of these factors and the implications for substance use treatments. At present, some evidence suggests that shame results in vulnerability to addiction to alcohol and illicit drugs, but there is little understanding of the mechanisms underlying this relationship. Similarly, there is an absence of evidence relating to the age at which shame-prone adolescents are most vulnerable to substance use. Few studies have investigated young people exclusively and, of the papers reviewed here, none has compared adolescents at different developmental stages. It may be that there is a point of 'greatest vulnerability' and, if so, there will be significant implications for the ways in which adolescents are educated about alcohol and illicit substances, and preventative measures are established. In addition, the majority of the studies used samples in which females were over-represented. Future work should attempt to redress the balance by investigating the shame-substance use relationship in young people of both sexes.

In this review, only papers relating to Western cultures were included. Although there was a clear rationale for this, given the prevalence of adolescent substance use in the UK and USA, there has been limited scope to investigate the shame-substance use relationship, or the meaning surrounding substance use, in social sub-groups. Although some studies used samples diverse in ethnicity and age, the present review has noted little that is relevant to constructs of class or religious belief. Although such factors have been investigated to only a limited degree, some research has attempted to improve our understanding of them (*Rastogi & Wadhwa, 2006*; *Sandberg, 2010*). Using substances may result in heightened shame only in specific groups; alternatively, certain social clusters may be more or less inclined toward substance use. It should not be presumed that the findings of this review can be applied to all groups without further exploration of salient factors.

## Conclusions and future directions

The papers reviewed here indicate that adverse early experiences are associated with the development of shame, that behaviours considered to be impulsive and maladaptive, such as substance use, are significantly associated with feeling shameful (*Abramowitz & Berenbaum, 2007*; *Dearing, Stuewig & Tangney, 2005*; *Edwall, Hoffmann & Harrison, 1989*), and that amongst college students alcohol use arises from a desire to avoid unwanted feelings of isolation and inadequacy (*Lashbrook, 2000*). The absence of literature relating

to shame and substance use indicates a significant gap in theoretical understanding. Nevertheless, a recent bio psychosocial theory of motivation postulates that behaviour is fundamentally unstable and constantly re-determined on the basis of myriad stimuli, impulses and inhibitory forces (*West & Brown, 2013*). Consequently, vulnerability to engaging in substance use is unlikely to be constant; rather, individuals may present with heightened or diminished vulnerability at specific time-points, contingent on a number of predisposing and precipitating factors, such as the factors related to self-concept suggested here (*West & Brown, 2013*). Recent findings from longitudinal research have concluded that, whereas guilt-proneness in childhood predicts reduced substance use in alter adolescence, proneness to shame results in heightened engagement in risky behaviours, including use of alcohol and other drugs, independent of the influence of socioeconomic status (*Stuewig et al., 2014*). The results indicate that children's moral and emotional styles appear to be established even in middle childhood, and that there are clear and significant implications for behaviours which commonly occur in early adulthood. Further research should seek to investigate the mechanisms underlying these relationships and to ascertain the point at which appropriate interventions should be provided.

### Funding

The authors declare there was no funding for this work.

### Competing Interests

Robert Patton is an Academic Editor for PeerJ.

### Author Contributions

- Masuma Rahim conceived and designed the experiments, performed the experiments, analyzed the data, wrote the paper, prepared figures and/or tables, reviewed drafts of the paper.
- Robert Patton wrote the paper, prepared figures and/or tables, reviewed drafts of the paper.

### Supplemental Information

Supplemental information for this article can be found online at http://dx.doi.org/10.7717/peerj.737#supplemental-information.

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
