# Peer review of "The association between shame and substance use in young people: a systematic review"

_PeerJ, doi:10.7717/peerj.737_

## Round 0.1 · original submission · Minor Revisions

The authors state this a 'review of the literature', but it doesn't appear to be truly the case (at least it is not a literature review) so the authors should change the title for clarity's sake.

·

Basic reporting

The paper appears to meet all relevant criteria.

Experimental design

The paper appears to meet all relevant criteria.

Validity of the findings

The Introduction and Discussion are strong and interesting. Very few papers were available, or selected for review, which weakens the Results section, although the relative paucity of papers identified is a valid finding in itself.

Additional comments

There are a number of typographical errors in your text. These really should not be there as they give the impression that the paper has not been carefully checked. There are also what appear to be redundancies in the way you describe the de-selection of papers, as well as a couple of sentences of which I query the meaning. I have indicated these in the pdf.

Reviewer 2 ·

Basic reporting

The manuscript “The association between shame and substance misuse in young people: A review of the literature” provides an overview of the relatively scant literature on shame as it may relate to substance abuse problems in youth. The introduction distinguishes shame from guilt and clarifies possible associations between these two concepts as well as describing the development of shame in both children and adolescents. The authors then discuss shame-proneness and its association to psychopathology as well as the relationship between shame and negative behaviors and cognitions. After providing statistics on adolescent substance abuse, the authors briefly discuss the correlation between shame and substance abuse, leading to the rationale for the paper. The investigators conducted a systematic literature review to assess the extent of empirical evidence related to the relationship between shame and substance abuse in this population.

Experimental design

One strength of this manuscript is that the authors provide a detailed outline of their search strategy and explain how they arrived at and retained six relevant empirical studies. A table is used to briefly describe for each of the studies the aim, measures, sample, and findings. The review discusses all of these studies in more detail. Lastly, the discussion summarizes the findings from the six studies used and discusses limitations including: the inconsistencies in conceptualizing shame and guilt, lack of using discrete methods to ask adolescents about substance abuse, a lack of diversity in the samples, and limited focus on the cognitive and emotional processes that might mediate the relationship between shame and substance abuse.

Validity of the findings

Overall, this paper is effective in summarizing what is known about shame and substance abuse summarizing available research paramount to the topic and pointing out inconsistencies. There are thoughtful critiques of the research studies (e.g., measure conflating proneness to shame with treatment motivation). However, there are several conceptual issues the authors may wish to address.

First, the terms substance use, misuse, and abuse are seemingly used interchangeably throughout the manuscript and could be defined better. For example, the rational for the current review is clearly stated on the bottom of page 3; however, it is prefaced with “substance misuse” and then talks about “substance abuse”. Second, the concept of shame might be more explicitly defined and further distinguished from guilt in that the latter involves our own actions but shame is often internalized because of the actions of others (e.g., family member’s drinking) or simply status (e.g., poverty). Shame might also be related to adverse childhood events that increase the risk for developing substance abuse problems. This reader was wondering how shame would relate to the larger literature on internalized stigma, and Corrigan’s work discussed latter may be helpful here.

Third, the section “The development of shame and young people” is almost entirely devoted to children and could benefit from including more information on adolescents if possible. Fourth, the section on substance use among adolescents provides statistics but does not appear well linked to the paper’s thesis. It would seem that the motives literature (i.e. escapism, negative affect drinking) could be linked to shame. There is a large literature associating drinking to cope or escape with alcohol problems in adolescents. One also wonders if shame is more likely related to excessive drinking than say marijuana use. Finally, the paper ends discussing gaps in the literature and including some research suggestions. The paper might have a stronger ending with a brief separate section entitled Conclusions and Future Directions.

Additional comments

Overall, informative and thoughtful paper on under-researched area.

A recent related paper by Jeffrey Stuewig et al looking at prospective association of shame proneness and drinking is available on line and coming out in Child Psychiatry Hum Dev
DOI 10.1007/s10578-014-0467. Perhaps the authors might wish to add this recent work in their discussion section.

Some possible writing suggestions include:
Abstract -might state adverse childhood events and particularly sexual abuse
Abstract -Rather than “may be” is a mechanism by which some individuals cope
Pg. 1 typo: “oif” should be “of” in first paragraph
Throughout paper, consider stating “alcohol and other drugs” to emphasize that alcohol is a drug
Second paragraph last sentence: briefly explain how moderate levels of shame can be beneficial
Both shame and guilt enable negative self-evaluation
Pg. 2 typo: “socialised” should be “socialized” in sentence “Differences in the ways males and females are socialized can result in females…” in second paragraph
Pg. 2, 3rd paragraph, last sentence: Discusses maladaptive coping strategies (e.g. depression, eating disorders, and suicide). Might substance abuse be included here? Also, “disroders” should be “disorders”
Pg. 3 Use APA style with reporting percentages (i.e. Spell out if first word in sentence).
Pg. 3 3rd paragraph there is typo: “that” should be “than” in the sentence “In particular, females seeking treatment for substance-related problems may face greater stigma that males…”
Pg. 4 typos: Presumably, ‘adolescen*’ should be ‘adolescent’ and ‘substanc*’ should be ‘substance’ In sentence “Articles were searched for using terms related to ‘adolescence’ e.g., ‘adolescen*’…”
Pg. 6 typo: “on” should be “of” in 3rd paragraph, in the sentence “Of their undergraduate sample, 7.3% had symptoms on problematic alcohol use and…”
Pg. 7 typo: “recognise” should be “recognize” in sentence “In their study of 188 16-24 year-olds entering treatment for moderately problematic substance abuse, Rosenkranz, Henderson, Muller, & Goodman (2012) found that those individuals who reported greater shame-proneness were more likely to recognise their substance…”
Pg. 7 typo: “internalised” should be “internalized” in sentence “It appears that early maltreatment and neglect can result in heightened shame-proneness, possibly as a result of the adverse experiences being internalised, and the…”
Pg. 9 typo: “programmes” should be “programs” in sentence “This review has not included studies which aim to treat substance misuse per se; although two studies did use participants engaged in treatment programmes…”
References APA Style Capitalize first word after a colon in title (e.g. Gilbert ref Body shame: A)

---

## Round 0.2 · accepted · Accept

The additional information is valuable to the reader and addresses reviewer concerns. We look forward to seeing this interesting report in publication shortly.